# Synthesis of Platinum Nanocrystals Dispersed on Nitrogen-Doped Hierarchically Porous Carbon with Enhanced Oxygen Reduction Reaction Activity and Durability

**DOI:** 10.3390/nano13030444

**Published:** 2023-01-21

**Authors:** Min Li, Feng Liu, Supeng Pei, Zongshang Zhou, Kai Niu, Jianbo Wu, Yongming Zhang

**Affiliations:** 1School of Chemistry and Chemical Engineering, Frontiers Science Center for Transformative Molecules, Center of Hydrogen Science, and Shanghai Key Lab of Electrical Insulation & Thermal Aging, Shanghai Jiao Tong University, Shanghai 200240, China; 2School of Chemical and Environmental Engineering, Shanghai Institute of Technology, Shanghai 201418, China; 3State Key Laboratory of Metal Matrix Composites, School of Materials Science and Engineering, Center of Hydrogen Science, Materials Genome Initiative Center, Future Materials Innovation Center, Zhangjiang Institute for Advanced Study, Shanghai Jiao Tong University, Shanghai 200240, China

**Keywords:** fuel cell, electrocatalyst, ion exchange, Pt-N-C bonding, oxygen reduction reaction

## Abstract

Platinum-based catalysts are widely used for efficient catalysis of the acidic oxygen reduction reaction (ORR). However, the agglomeration and leaching of metallic Pt nanoparticles limit the catalytic activity and durability of the catalysts and restrict their large-scale commercialization. Therefore, this study aimed to achieve a uniform distribution and strong anchoring of Pt nanoparticles on a carbon support and improve the ORR activity and durability of proton-exchange membrane fuel cells. Herein, we report on the facile one-pot synthesis of a novel ORR catalyst using metal–nitrogen–carbon (M–N–C) bonding, which is formed in situ during the ion exchange and pyrolysis processes. An ion-exchange resin was used as the carbon source containing R-N^+^(CH_3_)_3_ groups, which coordinate with PtCl_6_^2−^ to form nanosized Pt clusters confined within the macroporous framework. After pyrolysis, strong M-N-C bonds were formed, thereby preventing the leaching and aggregation of Pt nanoparticles. The as-synthesized Pt supported on the N-doped hierarchically porous carbon catalyst (Pt/NHPC-800) showed high specific activity (0.3 mA cm^−2^) and mass activity (0.165 A mg_Pt_^−1^), which are approximately 2.7 and 1.5 times higher than those of commercial Pt/C, respectively. The electrochemical surface area of Pt/NHPC-800 remained unchanged (~1% loss) after an accelerated durability test of 10,000 cycles. The mass activity loss after ADT of Pt/NHPC-800 was 18%, which is considerably lower than that of commercial Pt/C (43%). Thus, a novel ORR catalyst with highly accessible and homogeneously dispersed Pt-N-C sites, high activity, and durability was successfully prepared via one-pot synthesis. This facile and scalable synthesis strategy for high-efficiency catalysts guides the further synthesis of commercially available ORR catalysts.

## 1. Introduction

Proton-exchange membrane fuel cells (PEMFCs) have received significant attention as promising energy conversion devices for a wide range of applications, such as in vehicles, power tools, and backup power systems, owing to their high energy efficiency and eco-compatibility [1]. However, their low oxygen reduction reaction (ORR) efficiency, limited service time, and high cost are the major challenges that restrict their use in commercial applications [2,3,4,5,6,7]. Current platinum-based catalysts can efficiently catalyze acidic ORR [8], and although it has been demonstrated that platinum-based catalysts are optimal ORR catalysts [9], their high cost and low stability are still of great concern [10]. Many studies have been conducted to address these drawbacks by developing alternative materials based on modified Pt and/or other metals [11,12,13,14,15,16,17]. Nevertheless, the agglomeration and leaching of metallic Pt nanoparticles are major limitations affecting the catalytic activity and durability of the catalysts [18,19,20]. Therefore, the development of new catalyst systems is urgently required to improve the loading of Pt nanoparticles on support materials to achieve better distributional uniformity [21,22,23].

The adhesion between Pt nanoparticles and carbon supports significantly influences the stability of Pt catalysts. Because this interaction is usually weak, heteroatoms (such as N, P, S, F, etc.) with a complex orbital composition can be introduced as the anchoring points to achieve orbital-level interactions [24]. It was demonstrated that nitrogen-doped carbon supports can better stabilize metallic Pt nanocrystals [25]. Moreover, co-doping of carbon with a metal and nitrogen (M-N-C) is another promising approach for tuning the electronic and geometric properties of the catalysts, which, in turn, modifies the electronic configuration of Pt nanoparticles [26,27]. Such modification facilitates the adsorption and desorption of the O* species [26,27,28,29]. It was shown that the Pt-N-C moieties of Pt–N co-doped carbon are the most effective active sites because they directly absorb O_2_ and subsequently catalyze the degradation of O–O bonds in acidic media [30,31], resulting in superior catalytic activity toward the ORR and enhanced fuel cell stability [32,33,34].

Further, the agglomeration of Pt nanoparticles during high-temperature pyrolysis results in a dramatic reduction in the interparticle porosity and the number of active sites; this is detrimental to the ORR activity and durability of the catalyst [35]. Thus, to improve the performance of catalysts, novel carbon support materials are being developed [36,37,38,39,40,41]. To develop a novel Pt catalyst on a carbon support with M-N-C bonding and high porosity, the intrinsic chemical properties of ion clusters embedded in a polymer matrix were explored using nanoscale phase separation with a carbon and metal blend, which was subjected to pyrolysis. Using this novel approach, polymers with abundant R-N^+^(CH_3_)_3_ groups were used to pre-control the electrostatic interactions for specific metal ions, which simultaneously yielded N-anchoring sites and strong bonding for metal nanoparticles upon pyrolysis [42]. The introduction of metal ions by replacing OH^−^ in situ yielded a material system with multiscale phase separation, which resulted in the high-porosity catalyst blends upon pyrolysis. The small metal nanocrystals formed by this approach were evenly dispersed owing to geometric constraints [43,44,45,46].

This approach was applied in the current study to synthesize Pt nanoparticles supported on N-doped hierarchical porous carbon (Pt/NHPC) nanocomposites with enhanced ORR activity (Figure 1). A porous poly(styrene-divinylbenzene) (Q-PSDB) copolymer with strongly basic quaternary ammonium groups (R-N^+^(CH_3_)_3_) was used as the carbon support precursor owing to the strong ability of R-N^+^(CH_3_)_3_ to coordinate the Pt precursor (PtCl_6_^2−^). Thus, PtCl_6_^2−^ ions were anchored on the Q-PSDB support precursor in situ [47,48]. Subsequently, carbonization was performed at 800 °C to immobilize the Pt nanocrystals on the N-doped hierarchical porous carbon support (Pt/NHPC-800) to prevent agglomeration and peeling.

## 2. Materials and Methods

### 2.1. Materials

All compounds were of analytical grade and were used as received without further purification.

### 2.2. Preparation of Poly(Styrene-Divinylbenzene) (PSDB)

Gelatin (0.8 g, Aladdin, Shanghai, China) was dissolved in deionized (DI) water (200 mL) at 60 °C for 30 min, and subsequently, styrene (20.0 g, J&K Scientific, Beijing, China), divinylbenzene (2.4 g, J&K Scientific), benzoyl peroxide (0.3 g, Sigma-Aldrich, Shanghai, China), and polyvinyl alcohol (1.1 g, Adamas, Shanghai, China) were sequentially added to the aqueous phase under continuous stirring. The mixture was heated to 80 °C and stirred for 2 h, heated to 88 °C for 2 h, and then solidified at 95 °C for 8 h. Finally, the obtained PSDB balls were washed with DI water and dried for 12 h at 60 °C in a vacuum oven [49].

### 2.3. Preparation of Chloromethylated PSDB (C-PSDB): Chloromethylation Reaction

Anhydrous PSDB (10.0 g), 37% hydrochloric acid (200 mL, Sigma-Aldrich), paraformaldehyde (10.0 g, Macklin, Shanghai, China), and ZnCl2 (5.0 g, Aladdin) were mixed in a 500 mL flask, and the mixture was stirred for 10 min at 0 °C and then heated at 45 °C for 16 h. Subsequently, the formed C-PSDB was washed with DI water until a pH value of 7 was achieved, and then dried in a vacuum oven for 12 h at 60 °C [49].

### 2.4. Preparation of Q-PSDB: Quaternization Reaction

C-PSDB powder (2.0 g) and trimethylamine (20 mL, Energy Chemical, Shanghai, China) were mixed at 0 °C for 10 min in a 50 mL flask, heated for 20 min at 45 °C, and stirred overnight at 25 °C to form the quaternary ammonium product Q-PSDB. The formed Q-PSDB was filtered, washed with DI water until a pH of 7 was obtained, and dried for 24 h at 40 °C [49].

### 2.5. Pretreatment of Q-PSDB

The Q-PSDB resin was pretreated in three main steps to remove possible impurities. First, the resin (10.0 g) was stirred in 200 mL of DI water at 50 °C for 1 h to remove the impurities that were loosely adsorbed on the surface. This procedure was repeated three times. Subsequently, the resin was soaked in 200 mL of 1 M HCl solution for 6 h and then washed with DI water until a neutral pH was attained. For the hydroxylation of the strongly basic resin, the resin was immersed in 200 mL of 1 M NaOH solution for 6 h and then washed with DI water until a pH of 7 was achieved. Ultimately, the resin was dried at 50 °C under vacuum and ground to form a powder.

### 2.6. Synthesis of Pt/NHPC-T

In a typical synthesis, the pretreated and ground Q-PSDB resin (1 g) was immersed in 6.5 mL of H_2_PtCl_6_ (J&K Scientific, 99.9%) solution (1 g of H_2_PtCl_6_·6H_2_O dissolved in 100 mL of DI water) and 6.5 mL of DI water. The dispersion was stirred for 12 h and then dried in a vacuum oven at 50 °C for 24 h. The resulting powder (Pt@Resin precursor) was pyrolyzed at controlled temperatures (700, 800, and 900 °C) at a heating rate of 2 °C min^−1^ under an N_2_ atmosphere for 2 h, and the final products were named Pt/NHPC-700, Pt/NHPC-800, and Pt/NHPC-900, respectively.

### 2.7. Characterization Techniques

Transmission electron microscopy (TEM) measurements were carried out at an acceleration voltage of 200 kV using a TALOS F200X scanning transmission electron microscope (Thermo Fisher Scientific, USA) equipped with an energy-dispersive X-ray spectroscopy detector. For the preparation of the TEM samples, the catalyst dispersion was dropped onto a copper grid covered by an ultrathin carbon film. A field-emission scanning electron microscope (Nova NanoSEM 450, FEI, USA) was used to capture scanning electron microscopy (SEM) images. Powder X-ray diffraction (PXRD) patterns were recorded on a D8-Advance diffractometer (Bruker, Germany) using Cu Kα radiation. The Brunauer–Emmett–Teller (BET) surface area was determined by performing nitrogen adsorption–desorption measurements using an ASAP 2460 surface area and porosimetry analyzer (Micromeritics, USA). The pore size distribution was determined using the Barrett–Joyner–Halenda (BJH) method. X-ray photoelectron spectroscopy (XPS) results were obtained using an Axis Ultra DLD XPS system (Kratos Analytical, UK). An iCAP 7600 ICP-OES analyzer (Thermo Fisher Scientific, USA) was used to perform the inductively coupled plasma-atomic emission spectroscopy (ICP-AES). Raman spectra were recorded on a DXR Raman microscope (Thermo Fisher Scientific, USA) with an excitation wavelength of 532 nm using an Ar laser source.

### 2.8. Electrochemical Measurements

An Autolab PGSTAT302N electrochemical workstation (Metrohm, the Netherlands) was used for the electrochemical experiments. For a conventional three-electrode cell configuration, a glassy carbon electrode (GCE; diameter = 3 mm; area = 0.07065 cm^2^) was used as the working electrode, Ag/AgCl as the reference electrode, and a platinum wire as the counter electrode. For the electrochemical measurements, 1 mg of the catalyst was added to a 1 mL solution of Nafion (5%), ethanol, and water at a volume ratio of 0.025:1:4. The mixture underwent ultrasonic treatment for 30 min to form a homogeneous ink with a concentration of 1 mg mL^−1^. Subsequently, a microsyringe was used to cast a suspension with a volume of 8 μL, corresponding to a catalyst loading of 113 μg cm^−1^, evenly onto the surface of the GCE, followed by a drying process in air. The hydrogen desorption charge was measured by integrating the cyclic voltammetry (CV) curve obtained at a scan rate of 50 mV s^−1^ in a N_2_-saturated 0.1 M solution of HClO_4_ at 25 °C to calculate the electrochemical surface area (ECSA), assuming that the value for the adsorption of a hydrogen monolayer was 0.21 mC cm^−1^ [50]. Linear sweep voltammetry (LSV) curves at various rotation speeds (400–2000 rpm) were obtained using a rotating disk electrode (RDE) at a potential scan rate of 10 mV s^−1^ in an O_2_-saturated aqueous 0.1 M solution of HClO_4_. Nitrogen-saturated electrolytes were used for CV tests to exclude O_2_ from the electrolytes to enable accurate determination of the ECSA. In addition, the LSV curves were obtained using oxygen-saturated electrolytes to ensure full contact between Pt and O_2_ to optimize the ORR performance. The Ag/AgCl electrode potential (*E*) was calibrated relative to the reversible hydrogen electrode (RHE), i.e., *E*_RHE_ = *E*_Ag/AgCl_ + 0.21 V + 0.059 × PH [50].

### 2.9. Kinetic Current (J_K_) Calculation

The Koutecky–Levich equation was used to calculate the kinetic current (*J*_K_), as expressed by Equations (1) and (2):(1)$$1/\left|J\right|=1/\left|J_{K}\right|+1/\left|J_{D}\right|=1/B\omega^{1/2}+1/J_{K}$$
(2)$$B=0.2nFC_{0}D_{0}^{2/3}v^{-1/6}$$
where *J* and *J*_*D*_ are the measured and diffusion-limiting currents, respectively, ω is the rotation speed in rpm, *n* is the electron transfer number, *F* is the Faraday constant (96,485 C mol^−1^), *C*_0_ is the bulk concentration of oxygen (1.26 × 10^−3^ mol L^−1^), *D*_0_ is the oxygen diffusion coefficient with a value of 1.93 × 10^−5^ cm^2^ s^−1^ in 0.1 M HClO_4_ solution, and *υ* is the kinetic viscosity (0.01 cm^2^ s^−1^) [27].

### 2.10. Specific Activity and Mass Activity

The electrochemical surface area (ECSA) was calculated based on the Pt mass, assuming that the value for the adsorption of a hydrogen monolayer was 0.21 mC cm^−2^ as expressed by Equation (3):(3)$$ECSA=\frac{S_{H}/V}{0.21m_{Pt}}$$
where *S_H_* is the area obtained by integrating the CV curves from 0.1 to 0.4 V, *V* is the scan rate, and *m*_*Pt*_ is the mass of Pt.

The specific activity was normalized by the electrochemical surface area (ECSA):(4)$$specific\;activity=\frac{J_{K}}{ECSA\times m_{Pt}}$$
where *J*_*K*_ is the kinetic current.

The mass activity was calculated based on *m*_*Pt*_.
(5)$$mass\;activity=\frac{J_{K}}{m_{Pt}}$$

## 3. Results and Discussion

### 3.1. Physical and Chemical Properties of Pt/NHPC-T

TEM was used to characterize the morphology of the synthesized catalysts, which revealed that Pt nanocrystals with a clear profile were homogeneously distributed on the carbon substrate, forming the hybrid structure of Pt/NHPC-800 (Figure 2a and Appendix A). The average grain size of the Pt nanoparticles was 2.7 nm (inset of Figure 2a). Compared with 20% commercial Pt/C (Johnson Matthey, London, UK), Pt/NHPC-800 showed higher dispersity without aggregation (Appendix A). In contrast, Pt/NHPC-700 had a wide particle size distribution, and Pt/NHPC-900 clearly showed agglomeration with an average particle size of 5.67 nm owing to the higher temperature (Appendix A). The PXRD pattern shows the typical face-centered cubic (fcc) structure of Pt crystallites with peak positions consistent with the standard powder diffraction file card JCPDS #04-0802 (Figure 2b), confirming the formation of a Pt nanocrystal structure. A grain size of 2.1 nm was calculated using the Debye–Scherrer equation. An independent Pt nanocrystal was analyzed using high-resolution TEM (Figure 2c), and the lattice spacings were measured to be 0.193 and 0.226 nm, corresponding to the {200} and {111} planes of the fcc Pt nanocrystal, respectively. The even distributions of Pt, C, N, and O elements throughout the support are shown by the TEM images and individual elemental maps (Figure 2d–h). The Pt content in Pt/NHPC-800 was determined to be approximately 16.5 wt%, as confirmed by ICP-AES and XPS (Appendix A).

The successful synthesis of highly dispersed and uniform Pt nanocrystals was due to the chemistry of the Q-PSDB polymer. First, macroporous Q-PSDB contains abundant R-N^+^(CH_3_)_3_ groups (Appendix A), which serve as trapping sites for PtCl_6_^2−^ ions during the in situ ion-exchange process, resulting in uniformly distributed Pt nanocrystals in the resin. Consequently, the surface chemistry and electrochemical equilibrium prevented the agglomeration of Pt nanoparticles during heat treatment due to the strong anchoring of Pt (Appendix A). Second, the hierarchical Q-PSDB resin leads to the hierarchically porous structure of the Pt/NHPC-800 catalyst (Figure 3b), which is particularly favorable for achieving significant mass transport during electrochemical reactions. Third, the support matrix is nitrogen-rich (Appendix A), and the strong interactions between Pt nanoparticles and pyridinic-N (noted as Pt-N-C) prevent Pt agglomeration and also act as an active catalytic site to further drive the ORR.

BET and Raman analyses were used to further explore the support structure of the Pt/NHPC-T catalysts (Figure 3). A combined type I/IV isotherm with a moderately steep hysteresis loop within the 0.4–0.6 range of the relative partial pressure is observed in Figure 3b, indicating that Pt/NHPC-800 has a hierarchical porosity structure with micro-, meso-, and macro-pores, which is consistent with the SEM results (Appendix A). These hierarchically porous structures are particularly favorable for the chemical adsorption and anchoring of Pt nanocrystals. Furthermore, the Pt/NHPC-800 catalyst had a total pore volume of 0.41 cm^3^ g^−1^ and a BET surface area of 538 m^2^ g^−1^, which are larger than those of Pt/C, Pt/NHPC-700 and Pt/NHPC-900 (Appendix A). The high-surface-area structures facilitate the dispersion of metal species owing to the increased surface-to-volume ratio [51]. Thus, Pt/NHPC-800 is expected to have superior electrochemical performance. Pt/NHPC-700 exhibited a predominantly microporous structure owing to the incomplete carbonization of the macroporous framework (Figure 3a). Pt/NHPC-900 had a similar hierarchically porous structure to Pt/NHPC-800 but a lower BET surface area owing to the shrinkage and collapse of the resin skeleton (Figure 3c). The lower surface area and lower total pore volume of commercial Pt/C were averse to the O_2_ transportation and anchoring of Pt nanoparticles (Figure 3d). The carbon chemistry of the hierarchically porous support materials was also studied by Raman spectroscopy. The peaks at 1338 and 1592 cm^−1^ correspond to the D and G bands of the carbon, respectively, and are observed for all samples (Appendix A) [52]. The D band reflects the defective carbon structure, and the G band represents the in-plane vibrations of the ideal sp^2^ carbon, which indicates a graphitic carbon. The peak area ratio of the D to G band (*I*_D_/*I*_G_) changed slightly depending on the carbonization temperature, from 2.04 for Pt/NHPC-700 to 1.89 for Pt/NHPC-800 and to 1.96 for Pt/NHPC-900. Thus, the highest degree of graphitization was obtained for the sample carbonized at 800 °C, which is beneficial for increasing the electrical conductivity and ORR activity [53]. The highest *I*_D_/*I*_G_ value for Pt/NHPC-700 confirmed the incomplete carbonization of the macroporous framework at 700 °C. The *I*_D_/*I*_G_ ratio increased at 900 °C, demonstrating a collapse of the carbonaceous skeleton.

The surface adhesion of the Pt nanocrystals on the N-doped carbon substrate was examined by XPS. The XPS spectrum exhibits evident C 1s, O 1s, N 1s, and Pt 4f peaks for Pt/NHPC-800 (Figure 4a), and the high-resolution XPS spectra showed two main peaks, which are attributed to the Pt 4f 5/2 and Pt 4f 7/2 orbitals (Figure 4b). In particular, three chemically distinct Pt species with binding energies (BEs) of 71.36 (4f 7/2) and 74.67 eV (4f 5/2) for metallic Pt(0), 72.08 (4f 7/2) and 75.56 eV (4f 5/2) for Pt(II) oxide, and 72.71 (4f 7/2) and 77.1 eV (4f 5/2) for Pt(Ⅳ) dioxide are observed in the spectrum [54,55]. Figure 4b shows a shift in the Pt 4f 7/2 peak to lower BEs (by approximately 0.5 eV) for Pt/NHPC-800, indicating a distinct modification of the Pt electronic configuration owing to electron donation from Pt to N, which confirms the interaction between Pt nanoparticles and pyridinic-N [56]. Similar electronic effects were reported for Pt/boron carbide catalysts, which exhibited a similar shift of the Pt 4f core level to higher BEs [57]. Therefore, the shift in the Pt 4f core level is likely a direct result of the interaction between Pt and N, which modulates the electronic structure of Pt and downshifts the d-band center, thus reducing the adsorption energy of O_2_ and facilitating the desorption of the O* intermediate products, consequently improving the ORR activity [26,28,58,59]. As shown in Figure 4c for the Pt@Resin precursor, the dominant peak of the Pt^4+^ species (72.52 eV) originated from the bonding between PtCl_6_^2−^ and quaternary ammonium-N^+^, and only a few unbound H_2_PtCl_6_ were observed. A large number of Pt(II) and Pt(IV) species were observed for Pt/NHPC-700, which confirms the limited reduction of PtCl_6_^2−^ owing to the relatively low pyrolysis temperature of 700 °C. Conversely, the Pt species in Pt/NHPC-900 were similar to those in Pt/NHPC-800, as evidenced by the position of the peaks and the determined ratios of various types of Pt species, owing to the stabilization of the nanocrystals from 800 to 900 °C. The contents of the three types of Pt, namely Pt(0), Pt(II), and Pt(IV), in the Pt@Resin, Pt/NHPC-700, Pt/NHPC-800, Pt/NHPC-900, and Pt/C samples are summarized in Appendix A. The metallic Pt(0) state predominated among all the Pt species with peak area ratios of approximately 62.8% and 66.8% observed in the Pt/NHPC-800 and Pt/NHPC-900 samples, respectively. These values are the highest among all samples, including Pt/C containing Pt(0) nanocrystals as the catalytic active sites.

The nitrogen-bonding configurations in Pt/NHPC processed at various temperatures were quantitatively analyzed by high-resolution XPS measurements. These results show that during the pyrolysis of the Pt@Resin precursors, Pt-N interactions occurred and N-C bonds were formed. The N 1s spectrum of Pt/NHPC-800 was deconvoluted into four peaks: 399.0 ± 0.2, 400.1 ± 0.2, 400.9 ± 0.2, and 402.5 ± 0.2 eV, corresponding to pyridinic-N, pyrrolic-N, graphitic-N, and oxidized-N, respectively, suggesting that nitrogen was successfully doped into the carbon skeleton [60] (Figure 4e). As shown in Figure 4d–g, Pt/NHPC-800 has the highest pyridinic-N content among the samples. As verified in many studies, in comparison to other N-bonding types, pyridinic-N is easier to coordinate with Pt to form Pt-N-C ORR active sites [25,26]. As a result, the presence of pyridinic-N induces a strong interaction between Pt nanocrystals and the N-doped substrates, preventing the leaching of Pt nanocrystals and improving durability. Furthermore, it has been shown that the carbon atoms close to pyridinic-N serve as catalytic active sites owing to their Lewis basicity, and O_2_ molecules are adsorbed at these sites during the initial step of the ORR [61]. Consequently, the highest content of pyridinic-N in Pt/NHPC-800 is conducive to the noticeable increase in the ORR activity.

### 3.2. Catalytic Performance of Pt/NHPC-T Catalysts

To choose an optimal Pt content for the ORR catalysts, we synthesized a series of x-Pt/NHPC-800 samples, which were prepared with the addition of x = 1.6 mL, 3.3 mL, 4.9 mL, 6.5 mL, and 8.0 mL of an aqueous H_2_PtCl_6_ solution with a concentration of 0.001 g mL^−1^. As shown in Appendix A, the as-prepared electrocatalyst prepared with 6.5 mL of the H_2_PtCl_6_ solution proved to be the best ORR catalyst, showing outstanding half-wave potential and electric current density at 0.9 V versus RHE in O_2_-saturated 0.1 M HClO_4_. Based on these results, Pt/NHPC-700 and -900 catalysts were prepared under the same conditions, and their ORR performance was measured in an O_2_-saturated 0.1 M HClO_4_ solution. To study the intrinsic kinetics of the catalysts, the ECSA values of Pt/NHPC-700, Pt/NHPC-800, and Pt/NHPC-900 based on the Pt mass were determined to be 16.9, 55.2, and 15.8 m^2^ g^−1^, respectively (Figure 5a and Appendix A). The corresponding ORR activities are listed in Appendix A. The half-wave potential of Pt/NHPC-800 was 0.878 V, which was the highest among all Pt/NHPC-T samples (0.746 V for Pt/NHPC-700 and 0.824 V for Pt/NHPC-900; Figure 5b and Appendix A). The mass activities of Pt/NHPC-700 and Pt/NHPC-900 were determined to be 0.013 and 0.038 A mg_Pt_^−1^, respectively, which were lower than that of Pt/NHPC-800 (0.165 A mg_Pt_^−1^). The Pt/NHPC-800 (0.300 mA cm^−2^) showed significantly higher specific activity than Pt/NHPC-700 (0.077 mA cm^−2^) and Pt/NHPC-900 (0.24 mA cm^−2^) (Figure 5c,d).

Figure 6a shows the CV curves of Pt/NHPC-800 compared with those of commercial Pt/C. The half-wave potential of Pt/NHPC-800 (0.878 V) was 22 mV higher than that of the commercial Pt/C catalyst (0.856 V), despite a lower ECSA value (55.2 m^2^ g^−1^) than that of Pt/C (96.0 m^2^ g^−1^) (Figure 6b). Figure 6c shows the ORR polarization curves of Pt/NHPC-800 at different rotating speeds. The fitted Koutecky–Levich curves (inset of Figure 6c) were highly linear and parallel. The slopes of the Koutecky–Levich plots were used to calculate the electron transfer number (*n*) of Pt/NHPC-800 (3.7), suggesting an approximate four-electron transfer process for ORR [52]. At 0.90 V, the mass activity of Pt/NHPC-800 (0.165 A mg_Pt_^−1^) was 1.5 times higher than that of the Pt/C catalyst (0.107 A mg_Pt_^−1^). The area-specific activity of Pt/NHPC-800 (0.300 mA cm^−2^) was 2.7 times higher than that of the commercial Pt/C (0.111 mA cm^−2^) (Figure 6d). As listed in Appendix A, the excellent mass activity, specific activity, and *E*_1/2_ values were superior to those of the commercial Pt/C catalyst.

The abovementioned results imply that the Pt/NHPC-800 catalyst has promising properties for ORR applications. Notably, a few key inherent advantages are thought to be the reason for its superior performance. (1) Well-tuned electronic and geometric properties of Pt by N atoms: many studies have illustrated that electronic effects originated from the strong interaction between metallic atoms, such as Pt, and pyridinic-N doped in the support (i.e., Pt-N-C), contributing to the change of the Pt surface electronic configuration (downshift of the d-band center), thereby reducing O_2_ adsorption [25,26,28,59]. (2) Use of the Q-PSDB resin with functional groups (electrostatic stabilization) and high porosity (steric stabilization): the uniform dispersion of the Pt nanocrystals, derived from the porous nature of Q-PSDB and bonding between the R-N^+^(CH_3_)_3_ and PtCl_6_^2−^ ions to capture and disperse PtCl_6_^2−^ in situ, which benefits to the surface chemistry and electrochemical equilibrium, prevent agglomeration and peeling. (3) A high degree of graphitization and large BET surface area with abundant micro-, meso-, and macropores facilitate the mass transport and diffusion of reactive species in the Pt/NHPC-800 electrode. (4) High contents of pyridinic-N: the carbon atoms close to pyridinic-N serve as additional catalytically active sites. Additionally, Pt/NHPC-800 has a great potential for application in membrane elec-trode assembly of the fuel cell. On the one hand, the abundant hierarchical pores with high surface area supply better oxygen accessibility to the Pt surface. On the other hand, the presence of the N-doped carbon support alters the interaction between the ionomer and the carbon support, facilitating proton accessibility, and changes the hy-drophobicity of carbon support, in turn affecting water management [62].

An accelerated durability test (ADT) was carried out to assess the durability of the catalysts using 10,000 potential cycles in an O_2_-saturated 0.1 M solution of HClO_4_ at 100 mV s^−1^ between 0.60 and 1.00 V (Figure 7), and CV cycles and ORR polarization curves were acquired. Compared with an ECSA loss of ~32% for Pt/C, the ECSA of Pt/NHPC-800 did not significantly change (~1% loss; Figure 7a,b). We hypothesize that the strong interactions between Pt and pyridinic-N, which prevent Pt agglomeration and peeling, may be responsible for the high ECSA retention of ~99%. The half-wave potential of the as-synthesized Pt/NHPC-800 negatively shifted by only 8 mV, revealing the excellent stability of the Pt nanocrystal catalyst during the ORR process (Figure 7c). In comparison, the half-wave potential of commercial Pt/C negatively shifted by more than 15 mV after 10,000 cycles under the same testing conditions (Figure 7d), indicating its poor stability relative to the Pt/NHPC-800 catalyst. The mass activity of Pt/NHPC-800 was 0.135 A mg_Pt_^−1^ after 10,000 potential cycles, whereas the corresponding mass activity of Pt/C was 0.061 A mg_Pt_^−1^ (Figure 7e). The mass activity losses of Pt/NHPC-800 and Pt/C were ~18% and ~43%, respectively. Meanwhile, the specific activity of the as-synthesized Pt/NHPC-800 exhibited a 17.6% loss from 0.300 to 0.247 mA cm^−2^ after 10,000 ADT cycles, while Pt/C exhibited a 17.1% loss from 0.111 to 0.092 mA cm^−2^ owing to the 32% higher ECSA loss for Pt/C (Figure 7f).

The high stability of Pt/NHPC-800 was confirmed by the TEM and SEM results. After ADT, the residual catalysts were removed from the electrodes by sonication and subjected to TEM and SEM characterization. As shown in Figure 8, after 10,000 ADT cycles, Pt/NHPC-800 exhibited a consistent and uniform dispersion of Pt with an average particle size of 2.96 nm, which was similar to that observed before the ADT. SEM maps are shown in Figure 8e,f. There is no obvious change in the morphology before and after ADT (Appendix A). The higher stability of the Pt/NHPC-800 catalyst compared to that of the commercial Pt/C is due to the strong interactions between the Pt nanoparticles and pyridinic-N (Pt-N-C), which prevents agglomeration and peeling. Furthermore, before the carbonization process, the macroporous channel-rich structure of Q-PSDB with R-N^+^(CH_3_)_3_ groups provides abundant and uniform sites that easily trap and disperse the PtCl_6_^2−^ precursor to form a relatively uniform distribution of Pt-N-C on the polymer.

High-resolution XPS spectra of Pt and N and XRD patterns were also obtained to confirm the structure of the Pt/NHPC catalyst after ADT (Figure 9 and Appendix A). Three chemically distinct Pt species with Bes of 71.45 (4f 7/2) and 74.79 eV (4f 5/2) for metallic Pt(0), 72.13 (4f 7/2) and 75.56 eV (4f 5/2) for Pt(II) oxide, and 72.68 (4f 7/2) and 77.23 eV (4f 5/2) for Pt(Ⅳ) dioxide are observed in the spectrum. There is nearly no shift in the peak position of the Pt^0^ species before and after ADT (Figure 9a). Moreover, the content of Pt^0^ was slightly reduced from 62.8 to 52% after ADT due to the incomplete desorption of the O* species from the Pt surface during the ORR (Appendix A). The N spectrum after ADT was fitted to four types of nitrogen species (Figure 9b). The pyridinic-N content decreased from 25% (before ADT) to 18% (after ADT), which is attributed to the ORR occurring in the N-doped support (Figure 9c). The XRD curve shows only slight changes after ADT, indicating that the Pt nanoparticles were stable during the ORR (Appendix A).

## 4. Conclusions

In summary, this study developed a convenient and efficient synthesis method for Pt/NHPC nanocomposite catalysts using an in situ electrostatic interaction strategy and heat treatment, resulting in uniformly immobilized and dispersed Pt nanocrystals on N-rich carbon supports. Compared with the commercial Pt/C catalyst, the as-synthesized Pt/NHPC-800 showed significantly enhanced electrocatalytic activity and durability. The outstanding ORR activity and stability were ascribed to a modified Pt electronic structure, uniform dispersion of the Pt nanocrystals owing to the resin with N-anchoring sites that strongly bond metal nanoparticles upon pyrolysis, high BET surface area, and hierarchical porosity with abundant active sites. The as-synthesized electrocatalysts have considerable potential for large-scale production owing to facile synthesis and have great prospects for use in membrane electrode assembly. However, compared with some alloy catalysts, the proposed catalysts still need performance improvements. This fundamental research offers a guideline for the novel and rational design of high-efficiency and commercially available catalysts for ORR.

## Figures and Tables

**Figure 1 nanomaterials-13-00444-f001:**
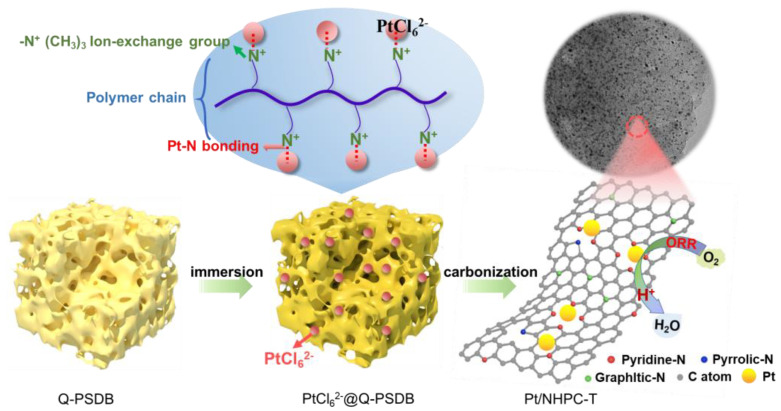
Synthesis of the nanocomposite-based catalyst consisting of Pt nanocrystals on a N-doped hierarchically porous carbon support (Pt/NHPC).

**Figure 2 nanomaterials-13-00444-f002:**
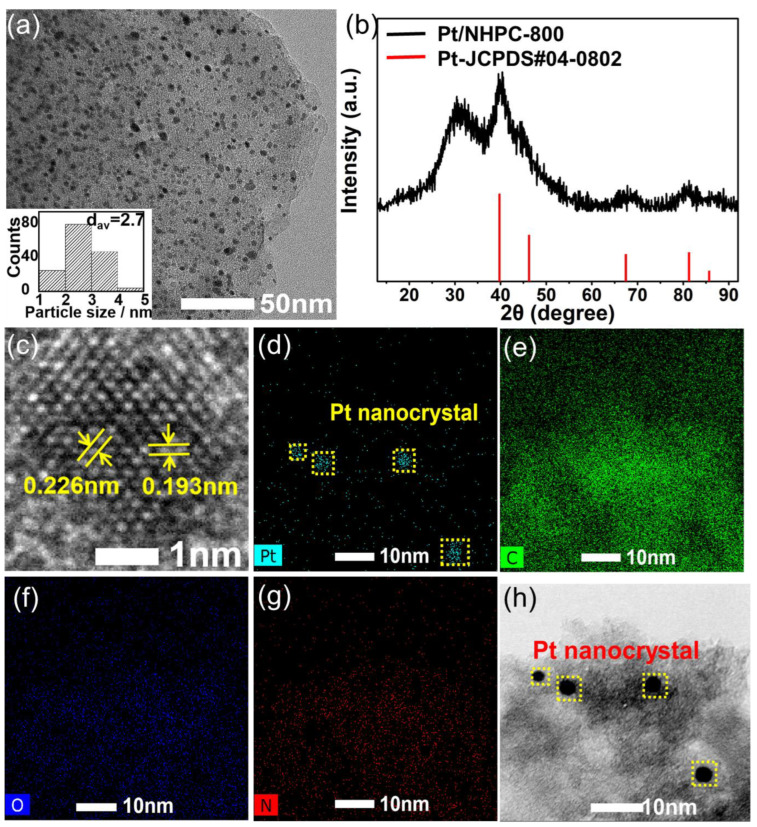
Characterization results for the as-synthesized Pt/NHPC-800. (**a**) Representative transmission electron microscopy (TEM) image (inset: particle size distribution). (**b**) X-ray diffraction pattern. (**c**) High-resolution TEM image of an individual Pt nanocrystal. Elemental maps of (**d**) Pt, (**e**) C, (**f**) O, and (**g**) N and the (**h**) corresponding TEM image.

**Figure 3 nanomaterials-13-00444-f003:**
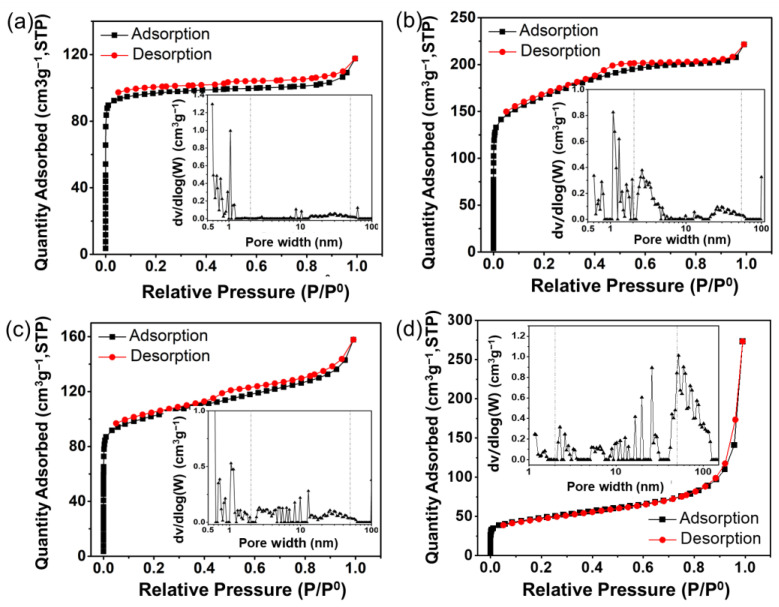
Nitrogen adsorption–desorption isotherms and (inset) pore width distributions of (**a**) Pt/NHPC–700, (**b**) Pt/NHPC–800, and (**c**) Pt/NHPC–900. (**d**) Pt/C.

**Figure 4 nanomaterials-13-00444-f004:**
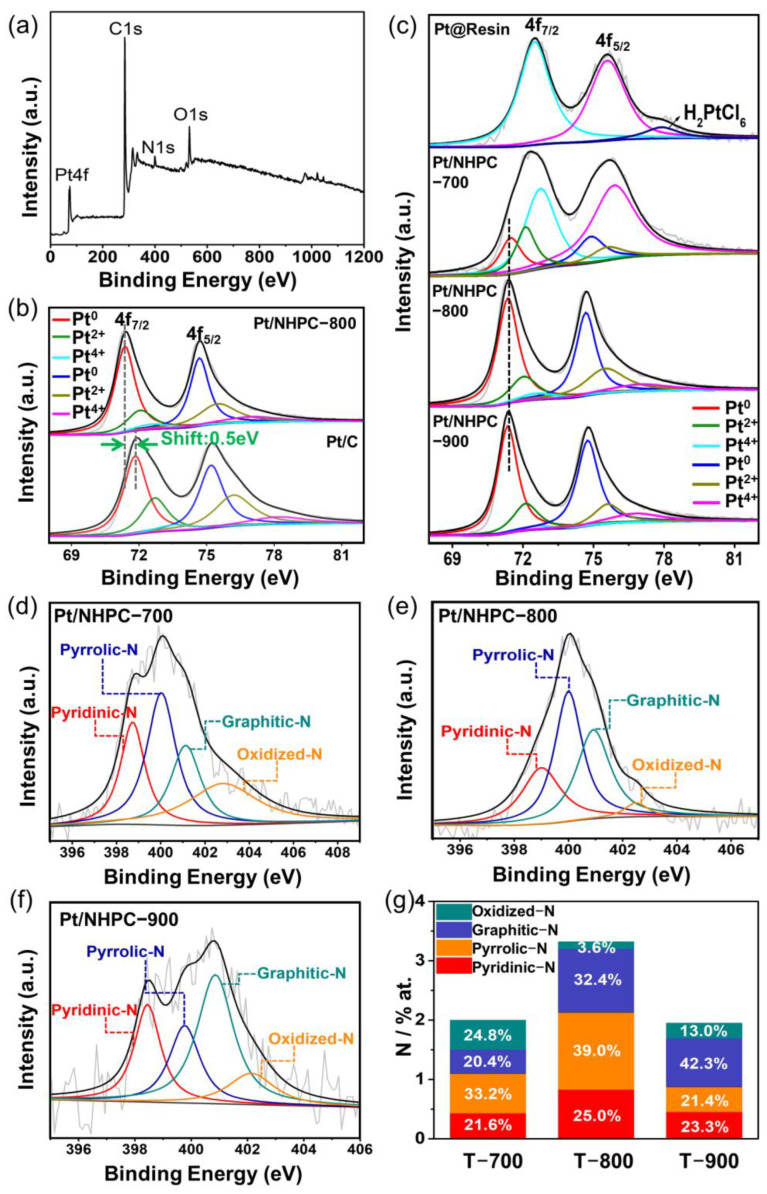
(**a**) X-ray photoelectron spectroscopy (XPS) survey spectrum of Pt/NHPC-800. (**b**) High-resolution XPS Pt 4f spectra of Pt/NHPC-800 and Pt/C. (**c**) High-resolution XPS Pt 4f spectra of the Pt@Resin precursor, Pt/NHPC-700, Pt/NHPC-800, and Pt/NHPC-900. High-resolution XPS N 1s spectra of (**d**) Pt/NHPC-700, (**e**) Pt/NHPC-800, and (**f**) Pt/NHPC-900. (**g**) Contents of the four types of nitrogen in the Pt/NHPC-*T* catalysts (*T* = 700, 800, and 900 °C).

**Figure 5 nanomaterials-13-00444-f005:**
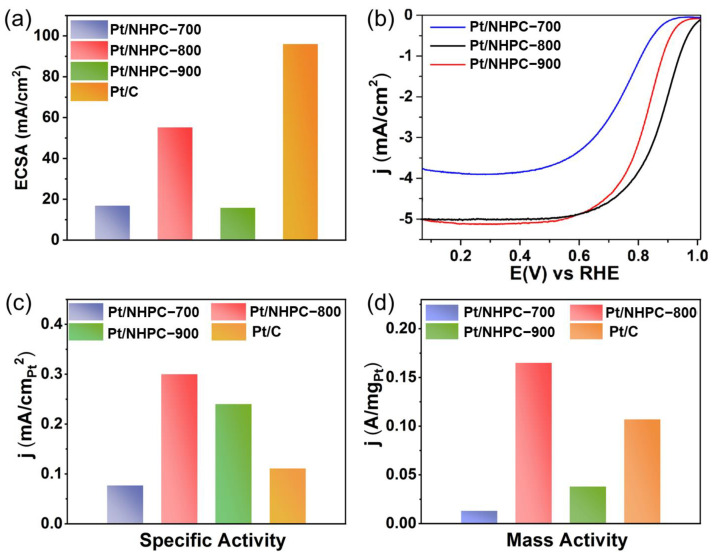
(**a**) Electrochemical surface areas (ECSAs) of all Pt/NHPC-*T* samples (*T* = 700, 800, and 900 °C) and Pt/C for comparison. (**b**) Linear sweep voltammetry curves of all Pt/NHPC-*T* samples obtained at a scan rate of 10 mV s^−1^ in O_2_-saturated 0.1 M HClO_4_ with a rotation speed of 1600 rpm. (**c**) Specific and (**d**) mass activities of all Pt/NHPC-*T* samples at 0.9 V versus RHE.

**Figure 6 nanomaterials-13-00444-f006:**
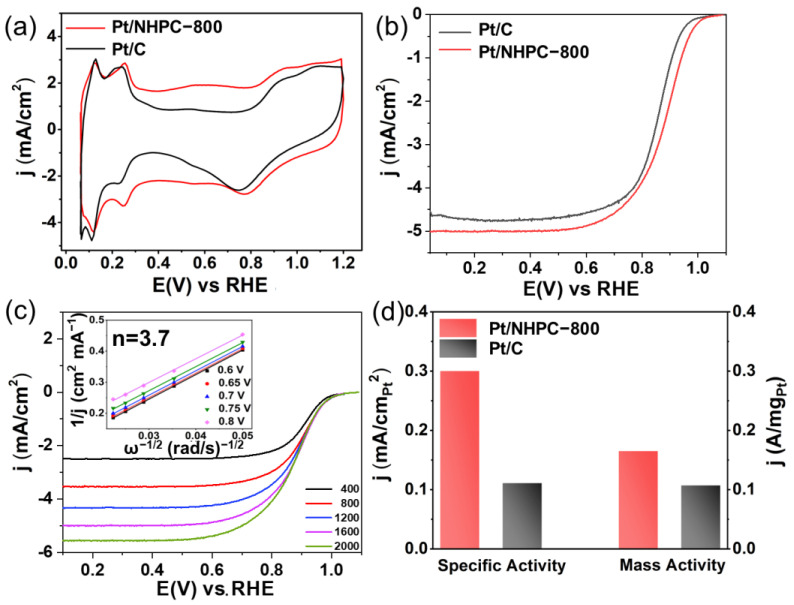
Representative oxygen reduction reaction (ORR) performance measurements. (**a**) Cyclic voltammetry data obtained at a scan rate of 50 mV s^−1^ in N_2_-saturated 0.1 M HClO_4_ and (**b**) LSV polarization curves at a rotation speed of 1600 rpm and a scan rate of 10 mV s^−1^ in O_2_-saturated 0.1 M HClO_4_ of Pt/NHPC-800 and commercial Pt/C. (**c**) ORR polarization curves of Pt/NHPC-800 at rotation rates of 400–2000 rpm (inset: Koutecky–Levich plots and the obtained electron transfer number, *n*). (**d**) Specific and mass activities of Pt/NHPC-800 and commercial Pt/C at 0.9 V vs. RHE.

**Figure 7 nanomaterials-13-00444-f007:**
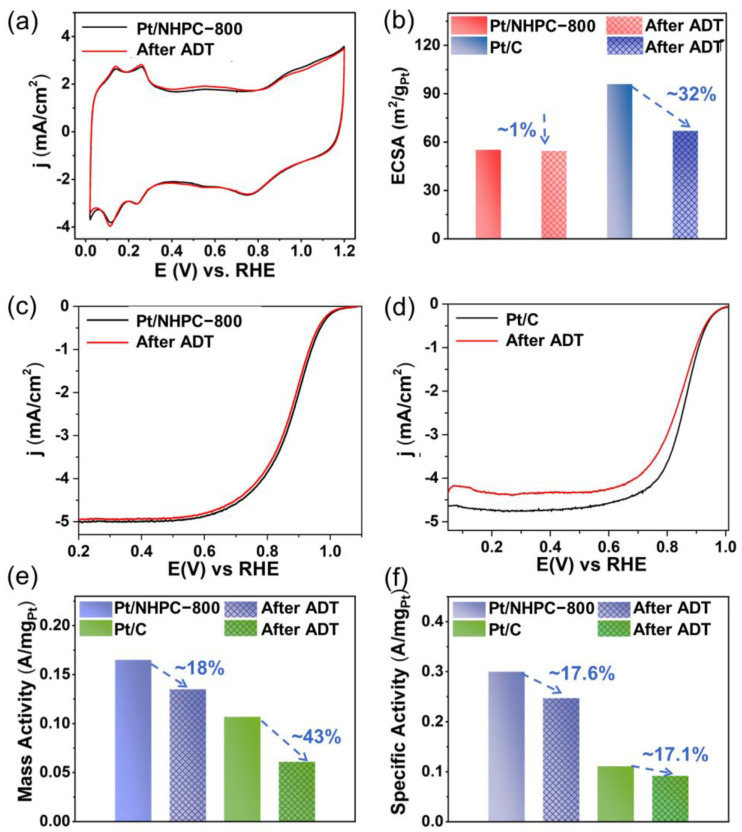
Representative ORR accelerated durability test (ADT) results. (**a**) CV curves of the long-term operational stability of Pt/NHPC-800 in N_2_-saturated 0.1 M HClO_4_ before and after 10,000 cycles from 0.6 to 1 V. (**b**) Comparison of ECSA values of Pt/NHPC-800 and commercial Pt/C before and after the ADT. RDE polarization curves of (**c**) Pt/NHPC-800 and (**d**) 20% commercial Pt/C before and after the ADT performed at a rotational speed of 1600 rpm and a scan rate of 10 mV s^−1^ from 0.6 to 1 V in O_2_-saturated 0.1 M HClO_4_. (**e**) Mass and (**f**) specific activities of Pt/NHPC-800 and 20% commercial Pt/C at 0.9 V vs. RHE before and after the ADT.

**Figure 8 nanomaterials-13-00444-f008:**
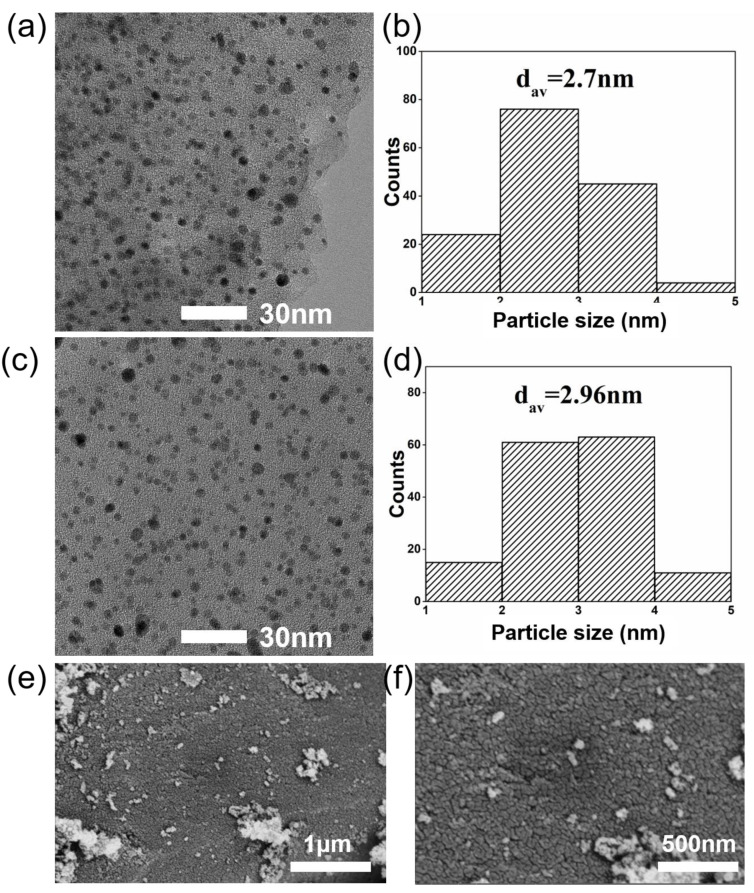
Representative (**a**) TEM image and (**b**) particle size distribution of as-synthesized Pt/NHPC-800. (**c**) TEM image and (**d**) particle size distribution of as-synthesized Pt/NHPC-800 after 10,000 ADT cycles. (**e**,**f**) SEM images after 10,000 ADT cycles of Pt/NHPC-800.

**Figure 9 nanomaterials-13-00444-f009:**
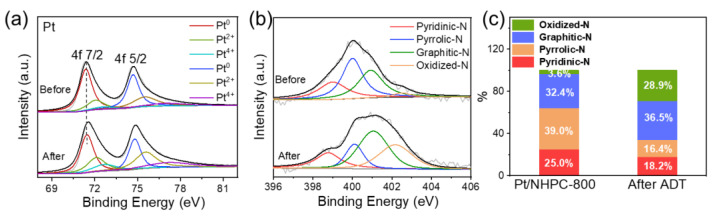
Analysis of Pt/NHPC-800 before and after ADT: (**a**) Pt XPS spectra; (**b**) N XPS spectra; (**c**) fractions of the N species.

## Data Availability

The data presented in this study are available within the article and Appendix A.

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
