# Peer review of "Synthesis of Platinum Nanocrystals Dispersed on Nitrogen-Doped Hierarchically Porous Carbon with Enhanced Oxygen Reduction Reaction Activity and Durability"

_nanomaterials, 2023, doi:10.3390/nano13030444_

Round 1

Reviewer 1 Report

Find the attached file.

Author Response

Dear Reviewer:

Thank you for the effort and comments concerning our manuscript entitled “Synthesis of platinum nanocrystals dispersed on nitrogen-doped hierarchically porous carbon with enhanced oxygen reduction reaction activity and durability” (nanomaterials-2141788). We took these comments seriously and made related revisions. Those comments are all valuable and helpful for improving the manuscript, and we wish this revision meet with your approval. Revised portions are marked in red in the manuscript. The main corrections in the paper and the point-by-point responses to the reviewers’ comments are summarized below

Reviewer 2 Report

The authors present the original preparation route for Pt/C catalyst with homogeneously dispersed Pt-N-C sites and a possible application as an oxygen reduction reaction (ORR) catalyst at the PEMFC cathode. There is no prior knowledge about the use of this specific catalyst preparation route making this highly important and novel work in the field. Thorough and versatile electrochemical and physical characterization is carried out together with the comparable or even better ORR performance than commercial Pt/C. The most important novelty is the high durability as ORR electrocatalyst when compared to the commercial Pt/C. For publication of the paper, some experimental information is missing and several RDE measurements results need to be critically reviewed and commented.

Major concerns:

1. The producer or origin of the chemicals throughout the section 2. Materials and Methods is missing.

2. The producer and product code for commercial 20wt% Pt type of Pt/C catalyst is missing. Also, the BET analyses with commercial Pt/C should be performed to compare the pore size distribution and BET surface area values with Pt/NHPC-800.

3. Figures 6b, 6c, 7c, S7b. The RDE polarization curve for Pt/NHPC-800 at most positive potential values starts at (negative) current density value, where no ORR should occur and the current should be 0 mA as the RDE measurements are background corrected. All the presented RDE polarization curves for other electrodes (including commercial Pt/C) start close to 0 mA as expected. Authors should double check their background current removal procedure for Pt/NHPC-800 RDE voltammetry curves. If necessary for solving the problem, at least the RDE voltammetry curve for Pt/NHPC-800 in Figure 6b should be measured and presented up to 1.1 V vs. RHE to see if the ORR curve will start at 0 mA as expected.

Minor concerns:

1. Numbers is superscript have problems throughout the manuscript, e.g. mg mL-1.

2. Page 2, line 45, recent important review article should be cited together with Refs. [10-15]: https://doi.org/10.1016/j.ijhydene.2020.08.215

3. Page 5, lines 170-172, the origin or reference for C0 and D0 values is missing.

4. Page 5, line 192, Table S2 should be Table S1.

5. Page 10, line 299, in Fig. S7b the same colours for the lines should be used as in S7a. Currently one cannot say which one is for 1.6 or 8.0 mL designation line.

Author Response

Dear Reviewers:

Thank you for the effort and comments concerning our manuscript entitled “Synthesis of platinum nanocrystals dispersed on nitrogen-doped hierarchically porous carbon with enhanced oxygen reduction reaction activity and durability” (nanomaterials-2141788). We took these comments seriously and made related revisions. Those comments are all valuable and helpful for improving the manuscript, and we wish this revision meet with your approval. Revised portions are marked in red in the manuscript. The main corrections in the paper and the point-by-point responses to the reviewers’ comments are summarized below

Reviewer 3 Report

In this work, a bifunctional Pt/NHPC-800 electrocatalyst is successfully prepared for oxygen reduction reaction activity which demonstrated 0.878 V half-wave potential (E 1/2 ) for ORR. Overall, the authors presented the experiment in detail and analyzed the results carefully. Although some results are appealing and the materials are well characterized, this current study is routine. It is necessary that the author enhance the innovation, and some insight mechanisms should be improved. Therefore, this manuscript could be suitable for this Journal after improvements. My detailed comments are as follows:

1. In the results and discussion section, the authors explain the results by showing Figures S1, S2….Table S1, Table S2….etc. If so, the authors should provide a supporting file to clarify the observed results based on the writing.

2. The authors are also focused on the specific activity and mass activity of the Pt/NHPC-800 for better ORR performance. Please clarify the ORR performance based on the specific activity and mass activity.

3. Pt is a noble metal element and is expensive compared to non-noble earth-abundant elements. Why did the authors use Pt instead of non-noble metals (Co, Ni, Fe, etc.) for the designing of Pt/NHPC-800? And what is the main objective of the work?

4. The authors claimed the electrocatalytic activity and kinetics of Pt/NHPC-800 for OER are greatly improved due to the presence of Pt and N-doped carbon. If so, the authors should provide related proof.

5. The XPS analysis of N 1s shows Pyrrolic N and Pyridinic nitrogen. Which nitrogen plays an active role in better ORR performance and why? With proper citation, XPS can be referenced from the "Carbon, Volume 179, July 2021, Pages 89-99.

6. There are many literature reports on the development of electrocatalysis, how does the performance level compare with previous literature? Similar research on the electrocatalyst can be cited in the appropriate positions for the reference of data presentation and explanation. Journal of Colloid and Interface Science, Volume 618, 15 July 2022, Pages 475-482 Composites Part B: Engineering, Volume 239, 15 June 2022, 109992

7. Why are nitrogen/oxygen-saturated electrolytes used for ORR performance evaluation? Add answer to the introduction section.

8. The surface reconstruction (FESEM, XPS, and XRD) after ORR was not discussed.

Author Response

(The authors gave the same response as above.)

Round 2

Reviewer 2 Report

The authors have very carefully and thoroughly revised the manuscript and answered to all of the reviewer’s questions and inquires. Therefore, the manuscript is now suitable for publication.

Author Response

Dear Reviewer:

We are very grateful to the reviewer for the positive efforts and valuable comments on the manuscript entitled “Synthesis of platinum nanocrystals dispersed on nitrogen-doped hierarchically porous carbon with enhanced oxygen reduction reaction activity and durability” (nanomaterials-2141788). The manuscript has been improved significantly. Many thanks to the reviewer for your approval of our manuscript.

Kind regards!

Reviewer 3 Report

1. In line 241 Fig. s6a, the s must be capitalized.

2. The conclusion must be made from the author's research work, so do not use references there. Reference numbers 57, 58, 59, and 60 need to manage in the appropriate place for the results and discussion.

Author Response

Dear Reviewer:

We thank the reviewer for the generally positive and valuable comments to improve the manuscript entitled “Synthesis of platinum nanocrystals dispersed on nitrogen-doped hierarchically porous carbon with enhanced oxygen reduction reaction activity and durability” (nanomaterials-2141788). We have addressed questions and concerns point by point raised by reviewer in the revised manuscript. The manuscript has been improved significantly. Revised portions are marked up using the “Track Changes” function in the manuscript.
